# The Impacts of COVID-19 on Healthcare Quality in Tertiary Medical Centers—A Retrospective Study on Data from Taiwan Clinical Performance Indicators System

**DOI:** 10.3390/ijerph19042278

**Published:** 2022-02-17

**Authors:** Shih-An Liu, Chieh-Liang Wu, I-Ju Chou, Pa-Chun Wang, Chia-Ling Hsu, Chia-Pei Chen

**Affiliations:** 1Center for Quality Management, Taichung Veterans General Hospital, Taichung 40705, Taiwan; saliu@vghtc.gov.tw (S.-A.L.); yrc731@vghtc.gov.tw (I.-J.C.); 2Department of Medical Research, China Medical University Hospital, China Medical University, Taichung 40604, Taiwan; 3Department of Medicine, School of Medicine, National Yang Ming Chiao Tung University, Taipei 11221, Taiwan; 4Department of Critical Care Medicine, Taichung Veterans General Hospital, Taichung 40705, Taiwan; cljeff.wu@gmail.com; 5Joint Commission of Taiwan, New Taipei City 22069, Taiwan; pachun.wang@jct.org.tw (P.-C.W.); linda.hsu@jct.org.tw (C.-L.H.)

**Keywords:** COVID-19, healthcare quality, medical center, unscheduled returns, quality indicators

## Abstract

To date, COVID-19 is by far the most impactful contagious disease of the 21st century and it has had a devastating effect on public health in countries around the globe. Elective medical services have declined markedly since the outbreak of the COVID-19 pandemic. Few studies have compared changes in healthcare quality before and during the outbreak of COVID-19 in Eastern Asian countries. We aimed to explore the impacts of COVID-19 on healthcare quality among medical centers in Taiwan. This was a retrospective study that collected anonymized data from the Taiwan Clinical Performance Indicator system, which was founded by the Joint Commission of Taiwan, an organization to promote, execute, and certify the nation’s healthcare quality policies. We explored quality indicators reported by more than three-quarters of medical centers in Taiwan from January 2019 to December 2020. The year 2019 was defined as the baseline period and 2020 was defined as the period after the start of the outbreak of COVID-19. Quality indicators from different regions were analyzed. Unscheduled returns of emergency patients within 72 h of their discharge, unscheduled returns of hospitalized patients within 14 days of their discharge, and unscheduled returns of surgical patients to the operating room during hospitalization all declined during the COVID-19 outbreak. Interestingly, the proportion of acute ischemic stroke patients receiving intravenous tissue-type plasminogen activator (IV-tPA) increased during outbreak of COVID-19. There were significant regional variations in healthcare quality indicators among medical centers in northern and middle/southern Taiwan. The outbreak of COVID-19 changed different patterns of healthcare systems. Although healthcare quality seemed to improve, further investigation is warranted to better understand whether those who were in need of returning to the emergency room or hospital were reluctant or were prevented from travel by the shelter-in-place policy.

## 1. Introduction

To date, COVID-19 is by far the most impactful contagious disease of the 21st century and it has had a devastating effect on public health in countries around the globe [1]. The COVID-19 pandemic has resulted in an extensive transformation of health care systems worldwide [2]. A previous report from a single institute in Thailand found that readmission rates for heart failure patients were significantly lower during the peak of the COVID-19 pandemic and were slightly increased during the post-peak period of the COVID-19 pandemic in those managed with telemedicine [3]. Another single institute study from the United Kingdom revealed that postoperative infection after emergency general surgery during the COVID-19 pandemic was similar to that of the pre-COVID-19 period [4]. In contrast, a study from Italy indicated that the overall surgical site infection rate was lower in the COVID-19 era when compared with that of the pre-COVID-19 period [5]. In addition, the surgical site infection rate after neurosurgery during the COVID-19 pandemic was lower when compared with that before the COVID-19 outbreak [6]. A UK-based multicenter retrospective national survey of patients who underwent foot and ankle surgery before, during, and after a national lockdown showed that the postoperative infection rate remained similar among these three periods [7]. However, another multicenter study examining administrative data provided by 67 German hospitals found that the 30-day readmission rate and in-hospital mortality rate in patients with heart failure during the COVID-19 outbreak were both higher than rates before the COVID-19 pandemic [2]. The first COVID-19 case in Taiwan was confirmed on 21 January 2020, and the number of confirmed cases had climbed to 447 by the end of June 2020 [8]. Taiwan activated the Central Epidemic Command Center immediately after the outbreak of COVID-19. In addition, Taiwan initiated control policies including border control, surveillance, quarantine, resource allocation, and face mask usage to prevent the spread of COVID-19 [8,9].

The Joint Commission of Taiwan (JCT) is an organization established in 1999 funding by the Ministry of Health and Welfare, Taiwan Hospital Association, Taiwan Non-government Hospitals and Clinics Association, and Taiwan Medical Association. JCT has qualified to become a professional accrediting organization that is certified by the International Society for Quality in Health Care (ISQua). JCT is also committed to promoting quality and patient safety concepts, the use of quality management tools, and healthcare professionals training in order to help administrators of healthcare organizations and professional healthcare providers to improve healthcare quality. The Taiwan Clinical Performance Indicator (TCPI) system was founded in 2011 by JCT. TCPI is a nonmandatory reporting system. In addition, the TCPI system provides various peers comparative data and dashboard such that hospitals could check their performance and identify problems in a timely manner. The system includes quality indicators from hospitals all around Taiwan. Almost all certified hospitals reported and recorded indicator data via the TCPI system [10]. Therefore, the TCPI system was well maintained and validated. There were, in total, 488 quality indicators included in the TCPI system. The quality indicators are divided into nine categories. A detailed description is listed in Table 1 [11].

Most of the aforementioned studies on healthcare quality during the outbreak of COVID-19 are from Western countries. To the authors’ knowledge, this is one of the pioneering studies in East Asian countries comparing changes in healthcare quality before and during the outbreak of COVID-19 pandemic. Therefore, the aim of the current study was to explore differences in healthcare quality before and during the outbreak of COVID-19 among medical centers in Taiwan. We also aimed to compare regional differences in healthcare quality in Taiwan.

## 2. Materials and Methods

This retrospective study was conducted in accordance with the Declaration of Helsinki and was approved by the institutional review board of Taichung Veterans General Hospital (protocol no: CW21266A). The requirement for patient consent was waived as all personal data in this secondary database were anonymized.

In the current study, data were retrieved from the TCPI system, which was founded by the JCT and included quality indicators from medical centers in Taiwan. We planned to explore all quality indicators from January 2019 to December 2020. The inclusion criteria of quality indicators were those reported by medical centers in the TCPI system. Quality indicators that were reported by less than three quarters of all medical centers were excluded (e.g., surgical site infection rate). A total of 8 quality indicators were included in the final analysis. We calculated the average value of all quality indicators. In the analysis, three months was used as one interval unit. As the first case of COVID-19 was reported in January, 2020 in Taiwan [8], 2019 served as the baseline period, whereas 2020 served as the period of the COVID-19 outbreak. We also divided hospitals based on their geographic location, i.e., the northern group and southern group, for ease of comparison. As almost half of the medical centers are located in northern Taiwan, we included medical centers in central and southern Taiwan into the southern group. Statistical Process Control (SPC) charts were drawn based on the location of medical centers. 

Continuous variables between northern and southern groups were compared with a Student’s *t*-test. Variables among subgroups (different quarters in 2019/2020) were analyzed with ANOVA and post hoc comparison was made using the Least Significant Difference (LSD) method. Statistical significance was defined as a *p*-value less than 0.05. All statistical analyses were performed with the Statistical Package for the Social Sciences (IBM SPSS version 22.0; International Business Machines Corp, New York, NY, USA).

## 3. Results

### 3.1. Quality Indicators for Final Analysis

A total of eight quality indicators fulfilled the inclusion criteria and were included in the final analysis. There were six process indicators (the number in parentheses represents medical centers that participated and the abbreviation code) including the following: (1) proportion of ST elevation myocardial infarction (STEMI) patients receiving primary percutaneous coronary intervention (PPCI) within 90 min after arrival at the emergency room (ER) (20, AMI07); (2) proportion of patients leaving the ER without complete treatment (17, EDL01); (3) unscheduled returns within 72 h for the emergency patients (17, EDR02); (4) unscheduled returns within 14 days for the hospitalized patients, excluding critical discharge against medical advice (DAMA) (18, HSUR01); (5) unscheduled returns to the operating room (OR) for the surgical patients during hospitalization (16, SCR01); (6) proportion of acute ischemic stroke patients receiving intravenous tissue-type plasminogen activator (IV-tPA) (19, STK03). The other two were outcome indicators: in-hospital mortality rate, excluding critical DAMA (19, HSM02), and mortality within 48 h for surgical patients, excluding critical DAMA (18, SCM02). A detailed description of the eight quality indicators is provided in Appendix A.

### 3.2. Quality Indicators before and during Outbreak of COVID-19

A comparison of quality indicators before and during the outbreak of COVID-19 revealed significant differences in unscheduled returns within 72 h for emergency patients, unscheduled returns within 14 days for hospitalized patients, unscheduled returns to the OR for surgical patients during hospitalization, and the proportion of acute ischemic stroke patients receiving IV-tPA. We used the average value of the entire year (2019 and 2020) for comparison. The data are presented in detail in Table 2.

### 3.3. Quality Indicators of Different Regions

When all variables were divided according to location of medical centers, there were significant differences between the northern group and southern group in AMI07 (93.3 ± 3.4% vs. 89.8 ± 4.1%, *p* = 0.002) (Figure 1A); EDL01 (2.42 ± 0.17% vs. 4.08 ± 0.30%, *p* < 0.001) (Figure 1B); EDR02 (2.30 ± 0.23% vs. 2.00 ± 0.19%, *p* < 0.001) (Figure 1C); HSM02 (2.26 ± 0.20% vs. 1.99 ± 0.16%, *p* < 0.001) (Figure 2A); HSUR01 (0.14 ± 0.01%vs. 0.11 ± 0.01%, *p* < 0.001) (Figure 2B); SCR01 (0.66 ± 0.11% vs. 0.73 ± 0.07%, *p* = 0.011) (Figure 3B); and STK03 (8.29 ± 1.65% vs. 7.01 ± 1.55%, *p* = 0.008) (Figure 1D). However, there was no significant difference between these two groups in SCM02 (0.93 ± 0.25‰ vs. 0.83 ± 0.29‰, *p* = 0.220) (Figure 3A). 

### 3.4. Quality Indicators Focusing on First Quarter of 2020

As the first case of COVID-19 was reported in January 2020 in Taiwan, the first quarter of 2020 was deemed to be the outbreak period of COVID-19. Three-month periods were used as units of time to compare variables within the same quarter (Q) between 2019 and 2020. We compared quality indicators including all centers in the first phase. There was a significant difference between Q1 of 2019 and Q1 of 2020 in only STK03 (6.42 ± 1.61% vs. 9.20 ± 1.91%, *p* = 0.004) (Figure 4). Although the in-hospital mortality rate, excluding critical DAMA, was the highest in Q1 of 2020, the rate was comparable with that in Q1 of 2019 (2.41 ± 0.17% vs. 2.23 ± 0.30%, *p* = 0.129). When we compared the variables between Q1 of 2019 and Q1 of 2020 in the northern group, there were significant differences in EDL01 (2.70 ± 0.22% vs. 2.29 ± 0.10%, *p* < 0.001); STK03 (7.34 ± 1.65% vs. 10.32 ± 2.01%, *p* = 0.031); and SCR01 (0.70 ± 0.09% vs. 0.57 ± 0.07%, *p* = 0.043). In addition, when we compared the variables between Q1 of 2019 and Q1 of 2020 in the southern group, there were significant differences in AMI07 (93.1 ± 2.3% vs. 84.3 ± 2.1%, *p* = 0.006), STK03 (5.51 ± 1.13% vs. 8.08 ± 1.14%, *p* = 0.029), and HSM02 (2.04 ± 0.22% vs. 2.26 ± 0.06%, *p* = 0.021).

## 4. Discussion

This study represents one of the first studies comparing healthcare indicators on a national level in east Asian countries. Elective medical services declined markedly during the outbreak of COVID-19 in Taiwan [12]. The same phenomenon was observed similarly all over the world [2,3,4,6,7]. People were less willing to visit a hospital for fear of COVID-19, which was known to be contagious disease at the outset of the outbreak [13]. This could explain the reduction in unscheduled returns within 72 h for emergency patients and unscheduled returns within 14 days for hospitalized patients during the outbreak of COVID-19 when compared with the pre-COVID-19 period. Unscheduled returns to the OR for surgical patients during hospitalization were also markedly reduced during the outbreak of COVID-19. We speculate that, due to the decline of elective procedures, surgeons might have had more time to deal with surgical patients, resulting in lower complication rates. On the other hand, the proportion of acute ischemic stroke patients receiving IV-tPA increased significantly during the outbreak of COVID-19. This might be explained by the fact that the emergency service volume was reduced after the outbreak of COVID-19; moreover, physicians were likely able to evaluate stroke patients more promptly. The reason why there was no significant difference in the proportion of STEMI patients receiving PPCI within 90 min after arrival at the ER before and during the outbreak of COVID-19 might be because the abovementioned patients were approached emergently regardless of the risk of the contagious disease and, therefore, PPCI could not be delayed. 

Due to a strict quarantine policy and prompt use of Big Data analytics to conduct efficient contact tracing of potential COVID-19 cases, Taiwan had relatively low COVID-19 confirmed and mortality cases in the beginning of the COVID-19 outbreak [9,12]. This could explain why both the in-hospital mortality rate, excluding critical DAMA, and mortality within 48 h for surgical patients, excluding critical DAMA, remained similar before and during the outbreak of COVID-19. A study from Italy demonstrated that all-cause mortality rates increased during the outbreak of COVID-19, and other European countries showed a similar trend [14]. The difference between the abovementioned study and the current study might be explained by the dissimilar kinds of mortality rates that were observed. Our study explored in-hospital mortality without critical DAMA, whereas European studies analyzed all-cause mortality using government statistics. Another explanation could be the differences of confirmed COVID-19 cases among different countries. A higher in-hospital mortality rate was noted during the outbreak of COVID-19 in Burundi, a country in east-central Africa [15]. A possible explanation could be that the abovementioned study collected data from a single institute, while our study analyzed data from multiple institutes. Different cultural backgrounds and dissimilar healthcare systems might also partially account for the observed differences. 

It is interesting to note there were significant differences in quality indicators based on the location of medical centers. Healthcare inequity exists worldwide as well as in Taiwan [16]. A previous study found that a PPCI service in a tertiary center would only cover a minority of STEMI events and could cause geographical inequities [17]. Nearly half of the medical centers in Taiwan are located in the northern part of the country. This might explain why seven out of eight quality indicators were different based on the location of medical centers in Taiwan. A recent meta-analysis indicated that COVID-19 pandemic prolonged the door-to-balloon (D2B) time, which is the time from the arrival at the emergency department of patients with STEMI until a catheter guidewire crosses the culprit lesion in the cardiac catheterization room, and worsened clinical outcomes, especially in Eastern low-middle-income countries [18]. It was suggested that the delay in D2B was in the ER as it served as the gatekeeper for COVID-19 patients. However, as previously mentioned, the medical service of the ER was reduced during the outbreak of COVID-19 and the ER was not overwhelmed in Taiwan; this might be another reason for why the proportion of STEMI patients receiving PPCI within 90 min after arrival at the ER in Taiwan remained the same during the outbreak of COVID-19. 

The percentage of patients leaving the ER without complete evaluation or treatment ranged from 0.1% to 15% in the literature [19,20,21]. The reasons for patients leaving the ER without complete evaluation or treatment might be due to prolonged waiting time, resolved symptoms [20,21], feeling too ill to wait [20], self-referral to the hospital’s out-patient department [21], the patient lives close to the ER, and insurance status [22]. Our results showed that the average rate of patients leaving the ER without complete evaluation or treatment was around 3%, which was comparable with the abovementioned studies. However, it is interesting to note that there was a significant difference in the rate of patients leaving the ER without complete evaluation or treatment based on the location of medical centers. As previously mentioned, almost half of the medical centers in Taiwan are located in the northern Taiwan and the medical resources in northern Taiwan are sufficient. However, insufficient medical resources in central and southern Taiwan might result in overcrowding of the ER and prolonged waiting time, which can eventually result in a higher rate of patients leaving the ER without complete evaluation or treatment. Although the proportion of patients leaving ER without complete treatment did not differ before and during the outbreak of COVID-19, it was reduced during the outbreak of COVID-19 in northern Taiwan. Due to the fact that most of the confirmed COVID-19 cases were in northern Taiwan [23], the impact of COVID-19 was more severe in the north. It is reasonable to expect that patients visiting the ER were mostly in need of emergency management. As a result, patients in northern Taiwan tended to not leave the ER without complete evaluation or treatment. 

A previous study revealed that the rate of unscheduled returns within 72 h for emergency patients ranged from 0.4% to 18% [24]. The rate in the current study ranged from 2.11% to 2.33% during the 2-year period. The reasons for unscheduled returns included persistence or worsening of the symptom, new diagnosis due to an initial error in diagnosis, occurrence of an adverse effect [24], leaving the ER without complete evaluation or treatment, treatment error, or leaving against medical advice [25]. As previously mentioned, insufficient medical resources in central and southern Taiwan might have resulted in overcrowding of the ER and prolonged waiting times. Hence, patients in central and southern Taiwan probably sought other resources to resolve their medical problems rather than returning to the ER. This could explain why the rate of unscheduled returns within 72 h for emergency patients in southern Taiwan was lower than that of northern Taiwan.

The factors associated with unscheduled returns within 14 days for hospitalized patients include age, socio-economic status, emergency admission, procedures during the index admission, length of stay, previous admission, comorbidity, malnutrition, blood pressure, temperature within 24 h before discharge, and discharge with a nasogastric tube [26]. Our study found a significant reduction in unscheduled returns within 14 days for hospitalized patients during the outbreak of COVID-19. As previously mentioned, routine medical services reduced markedly during the outbreak of COVID-19. Healthcare workers might have had more time to handle hospitalized patients, and there may have been less pressure to discharge patients. In addition, people were hesitant to visit a hospital out of fear of contracting COVID-19. These considerations could explain why unscheduled returns within 14 days for hospitalized patients decreased during the outbreak of COVID-19.

Tejada Meza et al. found, in a study conducted in Spain, that there was a decrease in admitted patients due to stroke during the COVID-19 pandemic period. However, no significant change in the proportion of intravenous treatment was noted [27]. Our study showed a different result. The reason might be that Spain started a lockdown since mid-March in 2020, whereas Taiwan did not have a lockdown during that period. Other explanations could be the healthcare systems as well as cultural backgrounds of Taiwan and Spain being dissimilar. Previous studies indicated that there were significant differences in postoperative infections during the outbreak of COVID-19 [5,6]. However, as less than three-quarters of the medical centers in Taiwan uploaded their surgical site infection rate, we did not analyze these data.

This study has some limitations. First, we only investigated quality indicators from tertiary medical centers of Taiwan rather than all levels of hospitals. Secondly, medical centers in Taiwan reported dissimilar quality indicators to the TCPI system. Thus, we only collected quality indicators from more than three-quarters of the medical centers. Thirdly, another outbreak of COVID-19 occurred in May of 2021, and the data from 2021 are still being analyzed.

## 5. Conclusions

The outbreak of COVID-19 changed different patterns of healthcare systems. Unscheduled returns within 72 h for emergency patients as well as readmission within 14 days for hospitalized patients were reduced according to our study. Although healthcare quality seemed to improve, further investigation is warranted to determine whether those who need to return to the ER or hospital were hesitant or were prevented from travelling by the shelter-in-place policy.

## Figures and Tables

**Figure 1 ijerph-19-02278-f001:**
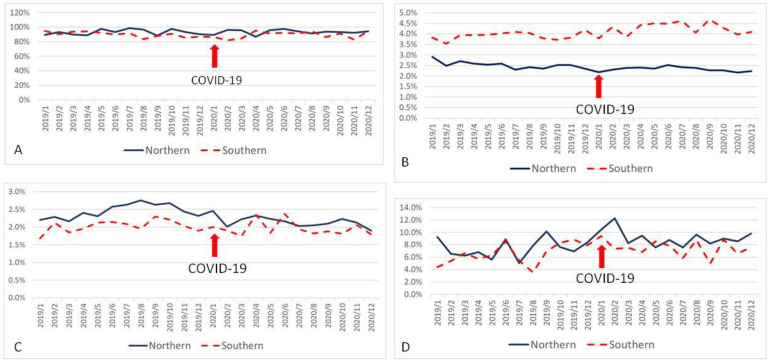
Control chart of emergency room-related quality indicators based on the location of medical centers (red arrow represents outbreak of COVID-19). (**A**): Proportion of STEMI patients receiving PPCI within 90 min after arrival at the emergency room. (**B**): Proportion of patients leaving the emergency room without complete treatment. (**C**): Unscheduled returns within 72 h for emergency patients. (**D**): Proportion of acute ischemic stroke patients receiving IV-tPA.

**Figure 2 ijerph-19-02278-f002:**
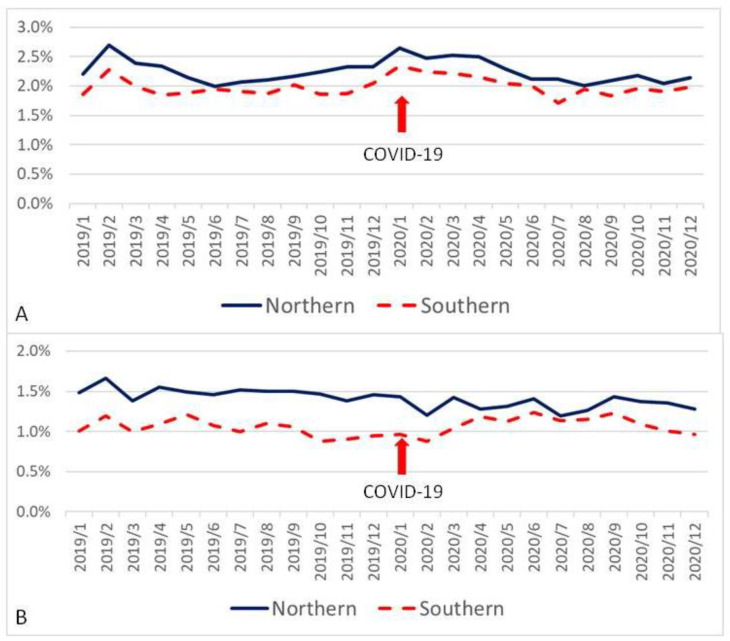
Control chart of hospitalization-related quality indicators based on the location of medical centers (red arrow represents outbreak of COVID-19). (**A**): In-hospital mortality rate excluding critical discharge against medical advice. (**B**): Unscheduled returns within 14 days for the hospitalized patients.

**Figure 3 ijerph-19-02278-f003:**
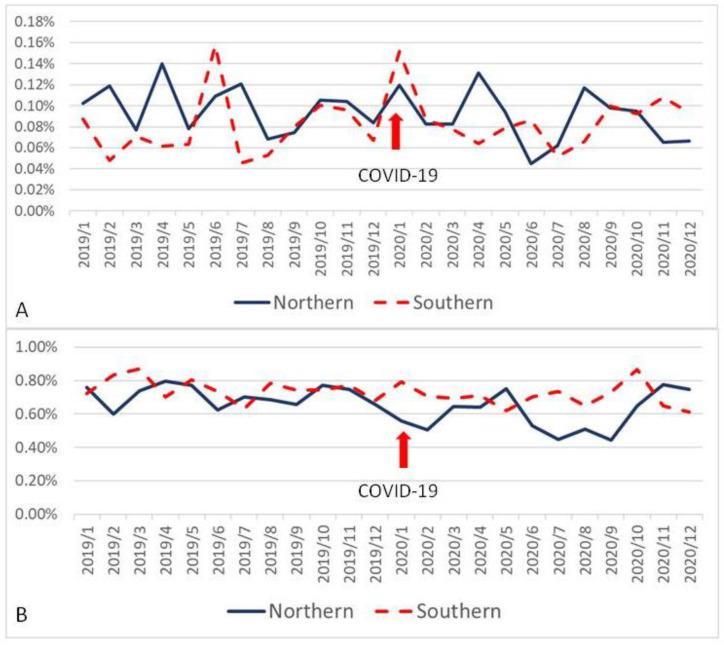
Control chart of surgery-related quality indicators based on the location of medical centers (red arrow represents outbreak of COVID-19). (**A**): Mortality within 48 h for surgical patients excluding critical discharge against medical advice. (**B**): Unscheduled returns to the OR for the surgical patients during hospitalization.

**Figure 4 ijerph-19-02278-f004:**
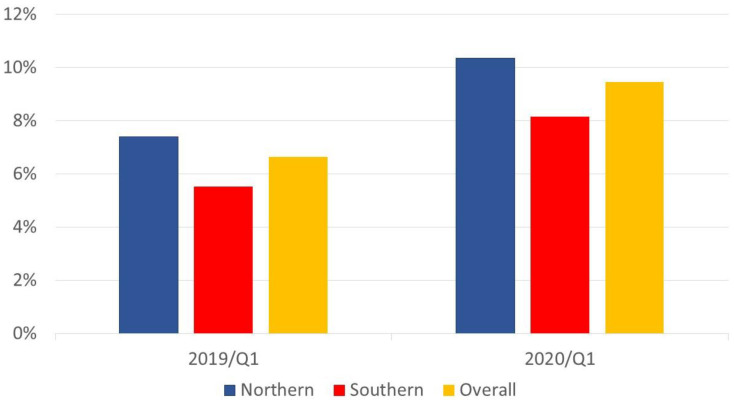
Comparison of the proportion of acute ischemic stroke patients receiving IV-tPA in Q1 2019 vs. Q1 2020.

**Table 1 ijerph-19-02278-t001:** Categories and numbers of quality indicators in TCPI system.

Category	Number of Quality Indicators
Overall performance	33
Intensive care unit-related	204
Surgery-related	43
Hospitalization-related	60
Pediatric	61
Obstetric	10
Emergency room-related	39
Acute myocardial infarction-related	16
Stroke-related	22
Total	488

**Table 2 ijerph-19-02278-t002:** Comparison of different indicators before and during the outbreak of COVID-19.

Quality Indicator	2019 (SD)	2020 (SD)	Difference (95% CI)	*p*-Value
AMI07	92.0 ± 2.4%	91.8 ± 2.4%	0.2 (−1.9~2.2)	0.896
EDL01	3.00 ± 0.11%	2.95 ± 0.15%	0.05 (−0.07~0.15)	0.431
EDR02	2.33 ± 0.17%	2.10 ± 0.14%	0.23 (0.04~0.15)	0.002 **
HSM02	2.12 ± 0.15%	2.16 ± 0.20%	−0.04 (−0.19~0.10)	0.546
HSUR01	1.32 ± 0.08%	1.23 ± 0.08%	0.09 (0.03~0.15)	0.008 **
SCM02	0.90 ± 0.20‰	0.88 ± 0.20‰	0.02 (−0.02~0.02)	0.838
SCR01	0.72 ± 0.05%	0.63 ± 0.07%	0.09 (0.04~0.14)	0.003 **
STK03	7.08 ± 1.21%	8.50 ± 1.08%	−1.42 (−2.40~−0.45)	0.006 **

Abbreviations: AMI07: Proportion of ST elevation myocardial infarction patients receiving primary percutaneous coronary intervention within 90 min after arrival at the ER; EDL01: proportion of patients leaving emergency room without complete treatment; EDR02: unscheduled returns within 72 h for emergency patients; HSM02: in-hospital mortality rate excluding critical discharge against medical advice; HSUR01: unscheduled returns within 14 days for hospitalized patients; SCM02: mortality within 48 h for surgical patients excluding critical discharge against medical advice; SCR01: unscheduled returns to the operating room for surgical patients during hospitalization; STK03: proportion of acute ischemic stroke patients receiving IV-tPA; SD: standard deviation; CI: confidence interval; Student’s *t*-test. ** *p* < 0.01

## Data Availability

Data are contained within the article or Appendix A.

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
