# Peer review of "The Impacts of COVID-19 on Healthcare Quality in Tertiary Medical Centers—A Retrospective Study on Data from Taiwan Clinical Performance Indicators System"

_ijerph, 2022, doi:10.3390/ijerph19042278_

Round 1

Reviewer 1 Report

The authors evaluate in this manuscript an increasingly interesting topic to health care systems globally nowadays, the impacts of COVID-19 pandemic on healthcare quality and relevantly the pandemic effects on quality indicators throughout comparison of different variables before and during the COVID-19 pandemic at a national level among different regions in Taiwan. 

The authors approached the topic orderly after a well-detailed background description of the impact of COVID-19 on different aspects of healthcare settings especially those relating to emergency admission (STEMI patients receiving PCI within 90 minutes of arrival to ER, ischemic stroke patients receiving IV-tPA) and unscheduled returns (unscheduled returns to the emergency department after 72 hours of discharge and unscheduled returns of hospitalized patients within 14 days of discharge). Throughout the manuscript, the authors supported their analysis and focused on the novelty of their evaluation regarding east Asian countries.  

This new analysis of the available retrospective anonymized database to develop one of the first evaluations of such impact of covid-19 on healthcare system quality indicators in an east Asian country (in that case of Taiwan) was constructed and discussed considerably and supported with the suitable statistical evaluation. 

The dataset and material were retrospectively collected and anonymized. The study was conducted after an ethical concept has been concluded (protocol no: CW21266A) in accordance with the Declaration of Helsinki and was approved by the institutional review board of Taichung Veterans General Hospital. The requirement for patient consent was waived as all personal data in this secondary database were anonymized. 

The manuscript was supported with well-illustrated figures and detailed tables. I recommend adding additional figures and tables to help illustrate further details and flow of reading (please read in the following detailed section).

Methodically, the authors excluded quality indicators that were reported by less than three-quarters of all medical centers in Taiwan. Thus ending with a total of 8 quality indicators for the final analysis. Further inclusion criteria were not mentioned. An additional detailed statement regarding the exact inclusion criteria could be beneficial for the manuscript. The year2019 served as the baseline period and the year 2020 as the COVID-19 outbreak. The centers were also divided into northern and southern groups and were compared with the Student’s t-test. The authors used ANOVA and post-hoc comparison was made using (LSD) method for analyzing variables among subgroups. The statistical analysis was performed with IBM SPSS version 22.0

The author presented their results comprehensively following the aforementioned thematic analysis. A total of 8 quality indicators for final analysis. The quality indicators were compared before and after the outbreak of COVID-19. And lastly, quality indicators were evaluated and compared among the different regions of Taiwan (northern vs. southern group). Additionally, an extra-analysis focused on the first quarter of 2020 v.s. Q1 of 2019 showed interestingly a significant difference in terms of the proportion of acute ischemic stroke patients receiving IV-tPA. 

The discussion, which follows reemphasizes the results of the previously mentioned quality indicators comparisons. The authors concluded their discussions with a group of limitations of their study, which could be potential targets for futuristic studies (e.g. inclusion of primary medical center in further analysis of future studies, extending the period of analysis to include longer COVID-19 periods). 

I suggest however minor edits and revisions before publication in MDPI IJERPH: 

In general, the authors should: 

- Title: In light of the study description and methodology, I suggest rephrasing the title as follow (The Impacts of COVID-19 on Healthcare Quality– A retrospective study on data from Taiwan Clinical Performance Indicators System)

- I suggest adding a list of the most commonly used abbreviations in this manuscript for ease of understanding the different acronyms. Please add this either after the Keywords section or after the supplementary materials section. (e.g. COVID-19, TCPI, JCT, etc..)

Abstract:

p. 1, row 4 of Abstract: I suggest using the term (eastern Asian countries) instead of eastern countries to prevent confusion with other eastern countries (e.g. Eastern Europeans or eastern African).

 p. 1, row 6 of Abstract - Please add the adjective (anonymized) to the following (that collected anonymized data).

 p. 1, row 7 of Abstract- Please add a brief description in one or two sentences on the ( Joint Commission of Taiwan)

p. 1, row 10 of Abstract- No need for the word (also)

p. 1, row 11 of Abstract- Please use the following (unscheduled returns of emergency patients within 72 hours of their discharge) instead of (Unscheduled returns within 72 hours for emergency patients)

p. 1, row 12 of Abstract- Please use the following (unscheduled returns of hospitalized patients within 14 days of their discharge) instead of (Unscheduled returns within 14 days for hospitalized patients)

p. 1, row 13 of Abstract- Please use the following (unscheduled returns of surgical patients to the operating room during hospitalization) instead of (unscheduled returns to the operating room for the

surgical patients during hospitalization)

p. 1, row 13 of Abstract- Please use the word (Interestingly) instead of (In addition)

p. 1, row 14 of Abstract- I recommend rephrasing the sentence to the following (There were significant regional variations in healthcare quality indicators among medical centers in northern and those in middle/southern Taiwan)

Keywords:

I suggest adding the word (quality indicators) to the keywords.

Introduction

In this section, the authors described the impacts of COVID-19 on healthcare quality on a global and national level citing many recent good examples of studies comparing the effects of COVID-19 on the admission/readmission rates. The authors paved the way orderly to their study throughout the above-mentioned literature resources e.g. Seretis et al, Mangwani et al among others. Afterward, the authors introduced the concepts of Taiwan Clinical Performance Indicator (TCPI) and the Joint Commission of Taiwan (JCT). I recommend in this regard adding a brief description of the exact purpose and functionality of the TCPI and JCT.

Overall the authors presented a well-formed and supported introduction.

p. 1, row 6 of Introduction - Please use (were) instead of (was)

 p. 1, row 8 of Introduction - I recommend using the original term (postoperative infection), which is used in the cited article instead of (surgical site infection rate)

p. 2, row 6- I recommend using the word (Whereas,) or (However,) before the following (in another multicenter study)

p. 2, row 15-16- As mentioned above I strongly recommend in this regard adding a brief description of the exact purpose and functionality of the TCPI and JCT.

p. 2, row 20 to row 23- Please add a table of the categories of quality indicators according to sample size. This will help in summarizing the following relevant quality indicators.

p. 2, row 25 - Regarding the following statement (there are no studies in the literature comparing changes in healthcare quality before and during the outbreak of COVID-19 from Eastern countries). I would recommend using a less confirmative statement (To the authors' knowledge, this is one of the pioneering studies in east Asian countries comparing changes in healthcare quality before and during the outbreak of the COVID-19 pandemic)

Materials and methods:

In General well and detailed description of the dataset, ethic approval, quality indicator exclusion, groups and subgroups division, statistical analysis methodology, and statistical software package.

p. 2, row 7 of methods and materials - regarding the following statement (Quality indicators that were reported by less than three-quarters of all medical centers were excluded ) could you please ad an example of a quality indicator, which had been excluded from the study (e.g. their surgical site infection rate)

p. 2, row 8 of methods and materials - Please add a clear statement regarding the inclusion criteria in the study.

Results:

Very well presented and comprehensively analyzed. I would suggest here moving Table 1 (Comparison of different indicators before and during the outbreak of COVID-19) after the section of (3.3. Quality indicators of different regions).In this regard, it would be helpful to add a special chart or figure representing the comparison of the proportion of acute ischemic stroke patients receiving IV-tPA in Q1 2019 vs. Q1 2020 (3.4. Quality indicators focusing on the first quarter of 2020 )

Discussion

p. 8, row 1 of discussions - I recommend rephrasing the first sentence to a less confirmative statement e.g. (This study represents one of the first studies comparing healthcare indicators on a national level in east Asian countries) 

p. 8, row 4 of discussions - Please add the word similarly ( was observed similarly all over the world).

p. 8, row 5 of discussions - Regarding the statement (People were less willing to visit a hospital for fear of catching COVID-19) please rephrase this sentence to a less confirmative statement or add a supportive literature citation to support the above-mentioned statement. 

p. 8, row 12 of discussions - Please use (On the other hand) instead of (similarly)

p. 9, row 4 - I recommend rephrasing the sentence as follows (the abovementioned patients were approached emergently regardless of the risk of the contagious infectious disease and therefore was not possible to delay the PPCI)

p. 9, row 8 - please delete the typo (thy)

p. 9, row 33 - Please add a brief description of the door-to-balloon (D2B) time

p. 10, row 10 - Please use the word (symptom) instead of (problem)

p. 10, row 20 - I recommend using the term (socio-economic status) instead of (social determinants) 

p. 10, row 31 - I recommend using the following statement (Tejada Meza, Herbert et al. found in a study conducted in Spain that there……..)

p. 10, row 39 - Same as mentioned above I recommend using the original term (postoperative infection), which is used in the cited article instead of (surgical site infection rate)

Conclusions

p. 10, row 1 of conclusions - I would suggest rephrasing the sentence as follows (The outbreak of COVID-19 changed different patterns of healthcare systems)

p. 10, row 3 of conclusions - please add the following word after reduced (according to our study)

Author Response

The authors evaluate in this manuscript an increasingly interesting topic to health care systems globally nowadays, the impacts of COVID-19 pandemic on healthcare quality and relevantly the pandemic effects on quality indicators throughout comparison of different variables before and during the COVID-19 pandemic at a national level among different regions in Taiwan. 

The authors approached the topic orderly after a well-detailed background description of the impact of COVID-19 on different aspects of healthcare settings especially those relating to emergency admission (STEMI patients receiving PCI within 90 minutes of arrival to ER, ischemic stroke patients receiving IV-tPA) and unscheduled returns (unscheduled returns to the emergency department after 72 hours of discharge and unscheduled returns of hospitalized patients within 14 days of discharge). Throughout the manuscript, the authors supported their analysis and focused on the novelty of their evaluation regarding east Asian countries.  

This new analysis of the available retrospective anonymized database to develop one of the first evaluations of such impact of covid-19 on healthcare system quality indicators in an east Asian country (in that case of Taiwan) was constructed and discussed considerably and supported with the suitable statistical evaluation. 

The dataset and material were retrospectively collected and anonymized. The study was conducted after an ethical concept has been concluded (protocol no: CW21266A) in accordance with the Declaration of Helsinki and was approved by the institutional review board of Taichung Veterans General Hospital. The requirement for patient consent was waived as all personal data in this secondary database were anonymized. 

The manuscript was supported with well-illustrated figures and detailed tables. I recommend adding additional figures and tables to help illustrate further details and flow of reading (please read in the following detailed section).

Methodically, the authors excluded quality indicators that were reported by less than three-quarters of all medical centers in Taiwan. Thus ending with a total of 8 quality indicators for the final analysis. Further inclusion criteria were not mentioned. An additional detailed statement regarding the exact inclusion criteria could be beneficial for the manuscript. The year2019 served as the baseline period and the year 2020 as the COVID-19 outbreak. The centers were also divided into northern and southern groups and were compared with the Student’s t-test. The authors used ANOVA and post-hoc comparison was made using (LSD) method for analyzing variables among subgroups. The statistical analysis was performed with IBM SPSS version 22.0

The author presented their results comprehensively following the aforementioned thematic analysis. A total of 8 quality indicators for final analysis. The quality indicators were compared before and after the outbreak of COVID-19. And lastly, quality indicators were evaluated and compared among the different regions of Taiwan (northern vs. southern group). Additionally, an extra-analysis focused on the first quarter of 2020 v.s. Q1 of 2019 showed interestingly a significant difference in terms of the proportion of acute ischemic stroke patients receiving IV-tPA. 

The discussion, which follows reemphasizes the results of the previously mentioned quality indicators comparisons. The authors concluded their discussions with a group of limitations of their study, which could be potential targets for futuristic studies (e.g. inclusion of primary medical center in further analysis of future studies, extending the period of analysis to include longer COVID-19 periods). 

I suggest however minor edits and revisions before publication in MDPI IJERPH: 

In general, the authors should: 

- Title: In light of the study description and methodology, I suggest rephrasing the title as follow (The Impacts of COVID-19 on Healthcare Quality– A retrospective study on data from Taiwan Clinical Performance Indicators System)

Reply: Thanks for your suggestion. We have modified the title according to reviewer’s recommendation.

- I suggest adding a list of the most commonly used abbreviations in this manuscript for ease of understanding the different acronyms. Please add this either after the Keywords section or after the supplementary materials section. (e.g. COVID-19, TCPI, JCT, etc.)

Reply: We have added most commonly used abbreviations after the supplementary materials section in revised manuscript.

Abstract:

  1. 1, row 4 of Abstract: I suggest using the term (eastern Asian countries) instead of eastern countries to prevent confusion with other eastern countries (e.g. Eastern Europeans or eastern African).

Reply: Thanks for your suggestion. We’ll modified that sentence in revised manuscript.

  1. 1, row 6 of Abstract - Please add the adjective (anonymized) to the following (that collected anonymized data).

Reply: Thanks for your suggestion. We’ll add “anonymized” in revised manuscript.

  1. 1, row 7 of Abstract- Please add a brief description in one or two sentences on the (Joint Commission of Taiwan)

Reply: Thanks for your suggestion. We’ll add a brief description on the JCT in revised manuscript.

  1. 1, row 10 of Abstract- No need for the word (also)

Reply: Thanks for your suggestion. We’ll deleted that word in revised manuscript.

  1. 1, row 11 of Abstract- Please use the following (unscheduled returns of emergency patients within 72 hours of their discharge) instead of (Unscheduled returns within 72 hours for emergency patients)

Reply: Thanks for your suggestion. We’ll amend that sentence in revised manuscript.

  1. 1, row 12 of Abstract- Please use the following (unscheduled returns of hospitalized patients within 14 days of their discharge) instead of (Unscheduled returns within 14 days for hospitalized patients)

Reply: Thanks for your suggestion. We’ll modify that sentence in revised manuscript.

  1. 1, row 13 of Abstract- Please use the following (unscheduled returns of surgical patients to the operating room during hospitalization) instead of (unscheduled returns to the operating room for the surgical patients during hospitalization)

Reply: Thanks for your suggestion. We’ll adjust that sentence in revised manuscript.

  1. 1, row 13 of Abstract- Please use the word (Interestingly) instead of (In addition)

Reply: Thanks for your suggestion. We’ll modify that sentence according to reviewer’s recommendation in revised manuscript.

  1. 1, row 14 of Abstract- I recommend rephrasing the sentence to the following (There were significant regional variations in healthcare quality indicators among medical centers in northern and those in middle/southern Taiwan)

Reply: Thanks for your suggestion. We’ll rephrase that sentence in revised manuscript.

Keywords:

I suggest adding the word (quality indicators) to the keywords.

Reply: Thanks for your suggestion. We’ll add quality indicators to the keywords in revised manuscript.

Introduction

In this section, the authors described the impacts of COVID-19 on healthcare quality on a global and national level citing many recent good examples of studies comparing the effects of COVID-19 on the admission/readmission rates. The authors paved the way orderly to their study throughout the above-mentioned literature resources e.g. Seretis et al, Mangwani et al among others. Afterward, the authors introduced the concepts of Taiwan Clinical Performance Indicator (TCPI) and the Joint Commission of Taiwan (JCT). I recommend in this regard adding a brief description of the exact purpose and functionality of the TCPI and JCT.

Reply: We have added more description about the exact purpose and functionality of the TCPI and JCT.

Overall the authors presented a well-formed and supported introduction.

  1. 1, row 6 of Introduction - Please use (were) instead of (was)

Reply: Sorry for the mistake. We’ll change the word in revised manuscript.

  1. 1, row 8 of Introduction - I recommend using the original term (postoperative infection), which is used in the cited article instead of (surgical site infection rate)

Reply: Thanks for your suggestion. We’ll modify that sentence according to reviewer’s recommendation in revised manuscript.

  1. 2, row 6- I recommend using the word (Whereas,) or (However,) before the following (in another multicenter study)

Reply: Thanks for your suggestion. We’ll add that adverb in revised manuscript.

  1. 2, row 15-16- As mentioned above I strongly recommend in this regard adding a brief description of the exact purpose and functionality of the TCPI and JCT.

Reply: We have added more description about the exact purpose and functionality of the TCPI and JCT in revised manuscript.

  1. 2, row 20 to row 23- Please add a table of the categories of quality indicators according to sample size. This will help in summarizing the following relevant quality indicators.

Reply: We have added a table to describe the categories of quality indicators in revised manuscript.

  1. 2, row 25 - Regarding the following statement (there are no studies in the literature comparing changes in healthcare quality before and during the outbreak of COVID-19 from Eastern countries). I would recommend using a less confirmative statement (To the authors' knowledge, this is one of the pioneering studies in east Asian countries comparing changes in healthcare quality before and during the outbreak of the COVID-19 pandemic)

Reply: Thanks for your suggestion. We’ll modify that sentence according to reviewer’s recommendation in revised manuscript.

Materials and methods:

In General well and detailed description of the dataset, ethic approval, quality indicator exclusion, groups and subgroups division, statistical analysis methodology, and statistical software package.

  1. 2, row 7 of methods and materials - regarding the following statement (Quality indicators that were reported by less than three-quarters of all medical centers were excluded) could you please ad an example of a quality indicator, which had been excluded from the study (e.g. their surgical site infection rate)

Reply: We have added an example of quality indicator excluded from the study in revised manuscript.

  1. 2, row 8 of methods and materials - Please add a clear statement regarding the inclusion criteria in the study.

Reply: We have added inclusion criteria in the study in revised manuscript.

Results:

Very well presented and comprehensively analyzed. I would suggest here moving Table 1 (Comparison of different indicators before and during the outbreak of COVID-19) after the section of (3.3. Quality indicators of different regions).In this regard, it would be helpful to add a special chart or figure representing the comparison of the proportion of acute ischemic stroke patients receiving IV-tPA in Q1 2019 vs. Q1 2020 (3.4. Quality indicators focusing on the first quarter of 2020 )

Reply: As we added additional table according to reviewer’s suggestion, table 1 will change to table 2 and will be moved to the section after (3.3. Quality indicators of different regions). We’ll also added another chart to present the comparison of the proportion of acute ischemic stroke patients receiving IV-tPA in Q1 2019 vs. Q1 2020 in section 3.4.

Discussion

  1. 8, row 1 of discussions - I recommend rephrasing the first sentence to a less confirmative statement e.g. (This study represents one of the first studies comparing healthcare indicators on a national level in east Asian countries) 

Reply: Thanks for your suggestion. We’ll modify that sentence according to reviewer’s recommendation in revised manuscript.

  1. 8, row 4 of discussions - Please add the word similarly ( was observed similarly all over the world).

Reply: We’ll add that word in revised manuscript.

  1. 8, row 5 of discussions - Regarding the statement (People were less willing to visit a hospital for fear of catching COVID-19) please rephrase this sentence to a less confirmative statement or add a supportive literature citation to support the above-mentioned statement. 

Reply: We have rephrased the sentence and cited relevant literature in revised manuscript.

  1. 8, row 12 of discussions - Please use (On the other hand) instead of (similarly)

Reply: Thanks for your suggestion. We’ll modify the sentence in revised manuscript.

  1. 9, row 4 - I recommend rephrasing the sentence as follows (the abovementioned patients were approached emergently regardless of the risk of the contagious infectious disease and therefore was not possible to delay the PPCI)

Reply: Thanks for your suggestion. We’ll modify that sentence according to reviewer’s recommendation in revised manuscript.

  1. 9, row 8 - please delete the typo (thy)

Reply: Sorry for the mistake. It should be “why”. We’ll modify the sentence in revised manuscript.

  1. 9, row 33 - Please add a brief description of the door-to-balloon (D2B) time

Reply: We have added a brief description of D2B time in revised manuscript. (which is the time from the arrival at the emergency department of patients with STEMI until a catheter guidewire crosses the culprit lesion in the cardiac catheterization room)

  1. 10, row 10 - Please use the word (symptom) instead of (problem)

Reply: We’ll change the word in revised manuscript.

  1. 10, row 20 - I recommend using the term (socio-economic status) instead of (social determinants) 

Reply: We’ll modify the sentence in revised manuscript.

  1. 10, row 31 - I recommend using the following statement (Tejada Meza, Herbert et al. found in a study conducted in Spain that there……..)

Reply: Thanks for your suggestion, we’ll modify that sentence according to reviewer’s recommendation in revised manuscript.

  1. 10, row 39 - Same as mentioned above I recommend using the original term (postoperative infection), which is used in the cited article instead of (surgical site infection rate)

Reply: We’ll modify the sentence according to reviewer’s suggestion in revised manuscript.

Conclusions

  1. 10, row 1 of conclusions - I would suggest rephrasing the sentence as follows (The outbreak of COVID-19 changed different patterns of healthcare systems)

Reply: Thanks for your suggestion. We’ll change the sentence in revised manuscript.

  1. 10, row 3 of conclusions - please add the following word after reduced (according to our study)

Reply: We’ll amend the sentence in revised manuscript.

Reviewer 2 Report

The paper is related to a very interesting topic, i.e. the impacts of COVID-19 on Healthcare Quality in an Eastern country

However, the paper has many weaknesses, which require major revision.

Material and Methods

It is somewhat surprising that although the TCPI system provides 488 quality indicators only 8 quality indicators were included in the final analysis. The authors should clarify why there is such a drastic reduction in the number of indicators

3.2. Quality Indicators before and after outbreak of COVID-19

It is not clear for which period the average value of quality indicators was calculated: in other words, is the average for 2019 calculated throughout the whole 2019? In Materials and Methods they refer to “three months was used as one interval unit”

3.3. Quality indicators of different regions

Eight figures to show the control charts are excessive. I would recommend reducing them to 1/2 figures, divided into panels.

Do the differences reported in this paragraph refer to the two-year (2019-2020) averages for each indicator? However, Since the objective of the paper is to verify the impacts of COVID-19 on Healthcare Quality, it would be much more interesting to verify if the impacts of COVID-19 on Healthcare Quality was different between northern and southern hospitals. In other words: there was a difference between the northern and southern centers at baseline? how the quality of medical care has changed after Covid in northern and southern hospitals

3.4. Quality indicators focusing on first quarter of 2020

There was a significant difference between Q1 of 2019 and Q1 of 2020 in only STK03 (6.42 +/- 1.61 % vs. 9.20 +/- 1.91 %, p = 0.004). Is this result obtained by comparing the means between Q1 of 2019 and Q1 of 2020 including all centers (both North and South)? It is not clear.

Discussion

Data in results indicate that in-hospital mortality rate remained similar before and during the outbreak of COVID-19. What means: “This might explained be the reason why the in-hospital mortality rate rising in the first quarter of 2020.” This information contradicts the results.

Only in the conclusions, it is clear that the authors have selected only tertiary medical center, i.e. hospitals dedicated to specific sub-specialty care. But I think that efficiency of primary care, particularly in the Covid era, is associated with improved Healthcare Quality. This limitation must be highlighted already by the title

Minor revisions:

Please indicate the acronym for IV-tPA when you mention it for the first time

Supplementary table 1. ED is the acronym for?

Author Response

The paper is related to a very interesting topic, i.e. the impacts of COVID-19 on Healthcare Quality in an Eastern country

However, the paper has many weaknesses, which require major revision.

Material and Methods

It is somewhat surprising that although the TCPI system provides 488 quality indicators only 8 quality indicators were included in the final analysis. The authors should clarify why there is such a drastic reduction in the number of indicators

Reply: TCPI is a non-mandatory reporting system. Therefore, not all the quality indicators were reported by medical centers. In order to compare the real differences among medical centers, we have to exclude indicators that were reported by less than three quarters of all medical centers. We have added more description in revised manuscript.

 3.2. Quality Indicators before and after outbreak of COVID-19

It is not clear for which period the average value of quality indicators was calculated: in other words, is the average for 2019 calculated throughout the whole 2019? In Materials and Methods they refer to “three months was used as one interval unit”

Reply: The data in table 1 (now change to table 2) is the average for one year. For comparison of different quarter, the average for three months will be used. We’ll add more description in revised manuscript.

3.3. Quality indicators of different regions

Eight figures to show the control charts are excessive. I would recommend reducing them to 1/2 figures, divided into panels.

Reply: Thanks for your suggestion. We have divided quality indicators into “emergency room-related”, “hospitalization-related”, and “surgery-related”. The number of figures reduced from 8 to 4.

Do the differences reported in this paragraph refer to the two-year (2019-2020) averages for each indicator? However, Since the objective of the paper is to verify the impacts of COVID-19 on Healthcare Quality, it would be much more interesting to verify if the impacts of COVID-19 on Healthcare Quality was different between northern and southern hospitals. In other words: there was a difference between the northern and southern centers at baseline? how the quality of medical care has changed after Covid in northern and southern hospitals

Reply: We aimed to compare the difference before and during COVID-19 outbreak in the beginning of the study. However, we accidentally found that there was a baseline difference between northern and southern hospitals in some quality indicators. We already have discussed abovementioned findings in discussion section.

3.4. Quality indicators focusing on first quarter of 2020

There was a significant difference between Q1 of 2019 and Q1 of 2020 in only STK03 (6.42 +/- 1.61 % vs. 9.20 +/- 1.91 %, p = 0.004). Is this result obtained by comparing the means between Q1 of 2019 and Q1 of 2020 including all centers (both North and South)? It is not clear.

Reply: We compared all centers in this section. We’ll add more description in revised manuscript.

Discussion

Data in results indicate that in-hospital mortality rate remained similar before and during the outbreak of COVID-19. What means: “This might explained be the reason why the in-hospital mortality rate rising in the first quarter of 2020.” This information contradicts the results.

Reply: Thanks for your reminding. We’ll delete related sentences.

Only in the conclusions, it is clear that the authors have selected only tertiary medical center, i.e. hospitals dedicated to specific sub-specialty care. But I think that efficiency of primary care, particularly in the Covid era, is associated with improved Healthcare Quality. This limitation must be highlighted already by the title

Reply: Thanks for your reminding. We’ll add “in medical center” in the title.

Minor revisions:

Please indicate the acronym for IV-tPA when you mention it for the first time

Reply: IV-tPA is abbreviation for intravenous tissue-type plasminogen activator. We’ll mention it for the first time.

Supplementary table 1. ED is the acronym for?

Reply: Sorry for the mistake. We’ll change to ER (emergency room).

Round 2

Reviewer 2 Report

The paper has been improved according review suggestion. Authors must
change the title:
"Quality in Medical Centers" must become
"Quality in tertiary medical centers"

Author Response

Thanks for your suggestion. We'll modify the title according to reviewer's recommendation.
